# Development of the TrAnsparent ReportinG of observational studies Emulating a Target trial (TARGET) guideline

Harrison J Hansford ![ORCID],[1,2] Aidan G Cashin ![ORCID],[1,2] Matthew D Jones,[1,2] Sonja A Swanson,[3,4,5] Nazrul Islam,[6,7] Issa J Dahabreh,[4,5,8] Barbra A Dickerman,[4,5] Matthias Egger ![ORCID],[9,10,11] Xavier Garcia-Albeniz,[4,12] Robert M Golub,[13] Sara Lodi,[4,14] Margarita Moreno-Betancur,[15,16] Sallie-Anne Pearson,[17] Sebastian Schneeweiss,[18] Jonathan Sterne ![ORCID],[19,20,21] Melissa K Sharp,[22] Elizabeth A Stuart,[23] Miguel A Hernan,[4,5,8] Hopin Lee,[24,25] James H McAuley[1,2]

For numbered affiliations see end of article.

**Correspondence to**
Mr Harrison J Hansford;
h.hansford@unsw.edu.au

## ABSTRACT

**Background** Observational studies are increasingly used to inform health decision-making when randomised trials are not feasible, ethical or timely. The target trial approach provides a framework to help minimise common biases in observational studies that aim to estimate the causal effect of interventions. Incomplete reporting of studies using the target trial framework limits the ability for clinicians, researchers, patients and other decision-makers to appraise, synthesise and interpret findings to inform clinical and public health practice and policy. This paper describes the methods that we will use to develop the TrAnsparent ReportinG of observational studies Emulating a Target trial (TARGET) reporting guideline.

**Methods/design** The TARGET reporting guideline will be developed in five stages following recommended guidance. The first stage will identify target trial reporting practices by systematically reviewing published studies that explicitly emulated a target trial. The second stage will identify and refine items to be considered for inclusion in the TARGET guideline by consulting content experts using sequential online surveys. The third stage will prioritise and consolidate key items to be included in the TARGET guideline at an in-person consensus meeting of TARGET investigators. The fourth stage will produce and pilot-test both the TARGET guideline and explanation and elaboration document with relevant stakeholders. The fifth stage will disseminate the TARGET guideline and resources via journals, conferences and courses.

**Ethics and dissemination** Ethical approval for the survey has been attained (HC220536). The TARGET guideline will be disseminated widely in partnership with stakeholders to maximise adoption and improve reporting of these studies.

## STRENGTHS AND LIMITATIONS OF THIS STUDY

⇒ The TrAnsparent ReportinG of studies Emulating a Target trial (TARGET) reporting guideline will be developed according to recommendations for health research reporting guidelines.
⇒ The TARGET working group has been established to include stakeholders from a variety of backgrounds.
⇒ A comprehensive piloting phase may increase the usability and uptake of the reporting guideline.

## INTRODUCTION

Observational studies can provide evidence on the causal effects of interventions when it is not feasible, ethical or timely to conduct a relevant randomised trial. However, making causal inferences from observational data is challenging due to confounding and design-related biases such as selection bias and immortal time bias.[1 2] Design-related biases can be avoided using the target trial framework.[3 4] The framework involves the specification of the hypothetical randomised pragmatic trial—the target trial—that would ideally be conducted and how this trial might be emulated using observational data.[3 4] The two stages of the target trial framework are (1) specification of the target trial, and (2) emulation of the target trial.[3 4] Using observational data to mimic a randomised experiment was proposed in the mid 20th century,[5–8] and extended to time-varying treatments by Robins in 1986.[9]

The value of using the target trial framework to design the analysis of observational studies has been recognised by international regulatory bodies in the field of medicine and health,[10–14] and the framework underpins the widely-used ROBINS-I tool for assessing risk of bias in non-randomised studies of interventions.[15] Studies that are explicit in using the target trial framework have been published with increasing frequency in leading general medical and specialty journals.[16–23]

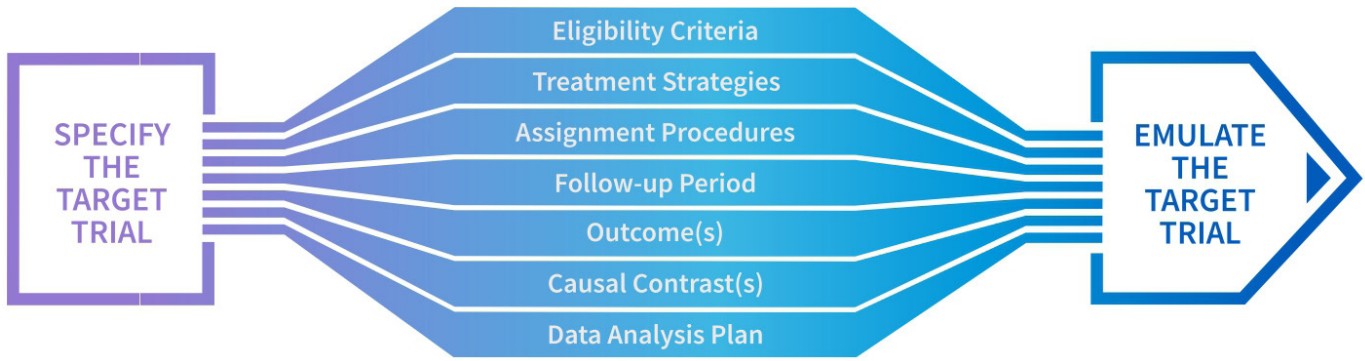

**Figure 1** Elements relevant to both the specification and emulation of the target trial described by Hernán and Robins.[3]

Application of the target trial framework requires the complete specification of the target trial protocol and its emulation (figure 1).[3] Hernán and Robins[3] provide a template for specifying a target trial and its emulation; however, there is currently no detailed guidance on reporting a study designed to emulate a target trial. Incomplete reporting of these studies limits the ability of clinicians, researchers, patients and other decision-makers to appraise and synthesise findings or interpret them to inform clinical and public health practice and policy. A reporting guideline that expands on the initial target trial emulation template[3] is needed to provide authors with comprehensive recommendations on how to completely and transparently report a study emulating a target trial.

To address this gap, we outline the processes and methods used to develop a reporting guideline for studies emulating a target trial—TARGET (TrAnsparent ReportinG of observational studies Emulating a Target trial).

## OBJECTIVE

The objective of the TARGET guideline is to provide guidance on the minimum set of items that should be reported to provide a clear and transparent account of observational studies that investigate the comparative effectiveness and safety of health interventions explicitly using the target trial framework.

## METHODS

We will develop the TARGET guideline in five stages following recommendations for the development of health research reporting guidelines (figure 2).[24] The start date for the study was in late 2022, with the planned end date in early 2025.

### TARGET working group

The TARGET working group is made up of the steering committee and project team (online supplemental material 1). The group was established to collate expertise on target trial emulation methodology, epidemiology, clinical trials, biostatistics, reporting guideline development

and knowledge of regulatory and journal editorial processes. The working group will oversee recruitment of participants for Stages 2 and 3 and contribute to writing and disseminating the guideline documents.

### Stage 1: identify current reporting practices

The systematic review aims to assess whether and how important items are reported by published studies explicitly emulating a target trial and whether reporting guidance (eg, Strengthening the Reporting of Observational Studies in Epidemiology[25]) was used. The protocol for this systematic review was registered on the Open Science Framework on 13 March 2022 (osf.io/uj56m).

### Databases, eligibility and search terms

We will search MEDLINE, EMBASE, PsycINFO and Science Citation Index for observational studies that stated in their methods that they explicitly emulated a target trial. We will exclude studies not written in English, not in the field of medicine and health, not conducted in humans or not observational designs. Many observational studies may implicitly use the framework of a randomised trial. However, to be included in this review studies must be explicit in their attempt to emulate a target trial (eg, stated 'target trial emulation' in the article). To identify eligible studies, we developed a literature search in collaboration with an expert librarian at the University of Oxford. Our approach used sensitive search terms including emulat*, target trial, observational data, real-world data, comparative effectiveness and causal inference, to try to capture all papers explicitly emulating a target trial. The complete search strategy is in online supplemental material 2. We will conduct forward citation tracking of selected seminal articles to maximise the chance of retrieving all relevant articles.[1 3 9 26 27] We will also include papers known to the authorship team. In duplicate, independent reviewers will conduct title, abstract and full text screening. We will resolve disagreements between reviewers through discussion.

### Data extraction

We will extract items regarded by the steering committee as potentially important for the reporting of a target trial emulation, including those outlined by Hernán and

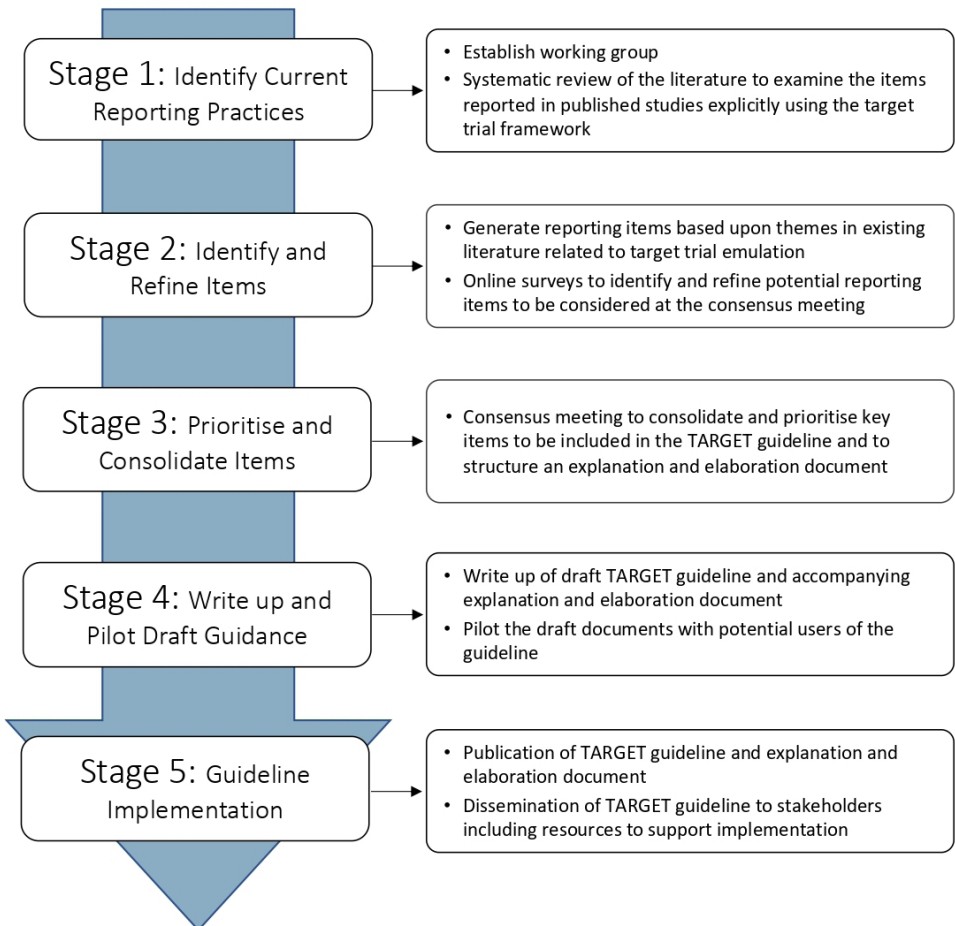

**Figure 2** Workflow for the development of the TARGET guideline. TARGET, TrAnsparent ReportinG of studies Emulating a Target trial.

Robins.[3] Two independent reviewers will extract information on study authors, year of publication, journal, subfield of medicine, study design, sample size, intervention, comparison group, outcomes assessed and whether the study was prospectively registered. We will extract items relevant to the methods and results of the target trial emulation, including whether and how all components of the protocol of the proposed target trial and how they were emulated, were specified (ie, eligibility criteria, treatment strategies, assignment procedures, follow-up period, outcome(s), causal contrast(s) and data analysis plan). We will enter data into a standardised data extraction form which two authors will pilot with a selection of included studies. We will resolve disagreements in data extraction between reviewers through discussion, or where necessary, consultation with a third reviewer.

### Data analysis
We will use R[28] for all data analyses. Categorical variables will be summarised using frequencies and percentages. Continuous variables will be summarised using mean and SD, or median and IQR, as appropriate.

### Outcomes of the systematic review
The systematic review will provide evidence on reporting in studies explicitly emulating a target trial. We acknowledge

that excluding studies not written in English and unpublished studies may cause potentially relevant articles to be excluded. The findings will inform the online surveys (Stage 2) and the consensus meeting (Stage 3). We will submit the findings of this review for publication and all data and code made publicly available.

### Stage 2: identify and refine items for the target guideline
We will conduct two online surveys to generate a list of candidate items that add detail to each of the protocol elements in figure 1.

### Ethics
Ethical approval has been obtained for the online surveys from the University of New South Wales Human Research Ethics Committee (HC220536).

### Selection of initial items
The steering group will develop a list of key items, informed by the systematic review (Stage 1) and the target trial framework described by Hernán and Robins,[3] thought important for the conduct and reporting target trial emulations (figure 1). Other potential sources of items include: published guidance for observational studies and randomised controlled trials, the ROBINS-I tool,[15]

and studies that describe items that may be important for the conduct or reporting of target trial emulations.

### Participants

Members of the TARGET working group (online supplemental material 1) will be invited to participate in the surveys.

### Procedure

We will host two online surveys using Research Electronic Data Capture (REDCap).[29 30] We will send each online survey via email to the participants. We will ask participants to rate the importance of each potential reporting item on a 9-point Likert scale (1, 'not important', to 9, 'critically important'). Participants will have the opportunity to provide suggestions or modifications to the wording of items as well as suggest additional items or make other comments.

In the second survey, we will send participants a summary of the results for each potential reporting item (mean scores and SD, median scores and IQRs and histograms), their own score for each item and any comments from participants on each item from the first survey. We will also present any new items and suggested modifications to items. We will then invite participants to rescore the importance of each item, and score any additional items, considering the aggregated ratings. Participants will have the opportunity to provide additional feedback on each item in the form of open ended responses.

### Analysis

Continuous variables will be summarised using mean and SD, or median and IQR, as appropriate. We will analyse the free-text responses from the first and second surveys using an inductive approach,[31] in which we will use reflexive thematic[31] analysis to identify, organise and generate codes and then identify themes found within the data set. Briefly, inductive coding is a process pooling common ideas without trying to fit ideas/codes into a pre-existing framework. These data will contribute to the creation of new items and modification of existing items to be included in the subsequent survey.

### Outcome of the online surveys

We will generate a preliminary list of items with corresponding ratings of importance to be considered in the TARGET guideline at the consensus meeting (Stage 3). We will also generate qualitative insights to guide item refinement and prioritisation in preparation for the consensus meeting.

### Stage 3: consolidate and prioritise key items to be included in the target guideline

A consensus meeting will finalise reporting items for the TARGET guideline.[24] The consensus meeting will follow suggested methods for developing reporting guidelines,[24] including guidance for consensus-based methods currently being developed which we will use if they become available.[32]

### Process

We will invite stakeholders identified by the working group to participate in a 2-day consensus meeting. The TARGET working group will ensure that the expertise of consensus meeting participants includes target trial emulation methodology, epidemiology, clinical trials, biostatistics, reporting guideline development and regulatory and journal editorial processes. Prior to the consensus meeting, the core team will provide attendees with evidence from the systematic review (Stage 1) and findings from the online surveys (Stage 2) including a draft of the items proposed for inclusion in the guideline. We will present the findings from Stage 1 and 2 at the consensus meeting. A member of the TARGET working group will facilitate a structured discussion on the rationale for including items from the online surveys. If there are disagreements, they will first be debated and, if disagreements remain, we will hold an anonymised vote to establish the importance of including the item in the guideline. For the anonymised vote, a simple majority will be sufficient to guide the inclusion/exclusion of an item. The meeting will conclude with discussion about the content and production of relevant documents (TARGET guideline, draft explanation and elaboration document) as well as strategies for dissemination and implementation. Following the conclusion of the consensus meeting, we will circulate a report on the outcome to the meeting participants for review and approval.

### Stage 4: development and piloting of the draft target guideline and explanation and elaboration document

Stage 4 involves drafting the TARGET guideline and accompanying explanation and elaboration document to ensure that the wording and content of the documents are clear, precise and suitable for all identified stakeholders. The purpose of the explanation and elaboration document is to explain each item by providing background information, a rationale and clear reporting examples from published target trial emulations. We will design the explanation and elaboration document to facilitate adherence to the TARGET guideline by clarifying the importance of each item, highlighting relevant reporting issues and providing examples to assist authors using the guideline. The consensus meeting participants may be asked to review and comment on the draft TARGET guideline and explanation and elaboration document.

We will evaluate the TARGET guideline by piloting the proposed guideline and the explanation and elaboration document with 20–30 expert methodologists and potential users of TARGET, identified from TARGET working group networks. We will ask participants to provide general feedback on accessibility and usability, and to identify possible reporting items that might have been overlooked. We will also ask for specific feedback about the utility and clarity of each TARGET item. We will collect data through online surveys, hosted by REDCap.[29 30] We will incorporate feedback from the piloting exercise into the final guideline and

explanation and elaboration document, as required. If suggested revisions are extensive, we will conduct a further round of piloting.

## Patient and public involvement

Potential users of this research include health researchers conducting observational analyses, regulatory bodies, public health and other health decision-makers. We aim to include relevant decision-makers in the piloting phase of the guideline development process to maximise the usefulness and uptake of the TARGET guideline. Participants in any stage of the guideline development will be informed of the results and final guidance.

## Stage 5: guideline implementation

The goal of the final stage of guideline development is to maximise reach and use of the TARGET guideline. The TARGET working group will guide the dissemination strategy with advice from consensus meeting participants. We aim to publish the TARGET guideline and the explanation and elaboration document and disseminate the findings through traditional and social media. We will engage journal editors and funding agencies to encourage TARGET guideline endorsement alongside other published reporting guidance. We will publicly host the TARGET guideline and explanation and elaboration paper, and any other relevant material on a TARGET website. We will index the guideline on the Enhancing the QUAlity and Transparency Of health Research Network website.[33][34] We will create online resources including infographics, blog posts and podcasts, which will be available on the TARGET website. We will share the TARGET guideline with authors in the field, and at relevant scientific conferences and methodological courses.

## Author affiliations

[1]School of Health Sciences, Faculty of Medicine and Health, UNSW Sydney, Sydney, New South Wales, Australia
[2]Centre for Pain IMPACT, Neuroscience Research Australia, Randwick, New South Wales, Australia
[3]Department of Epidemiology, University of Pittsburgh, Pittsburgh, Pennsylvania, USA
[4]CAUSALab, Harvard T.H. Chan School of Public Health, Boston, Massachusetts, USA
[5]Department of Epidemiology, Harvard T.H. Chan School of Public Health, Boston, Massachusetts, USA
[6]Oxford Population Health, Big Data Institute, University of Oxford, Oxford, UK
[7]Faculty of Medicine, University of Southampton, Southampton, UK
[8]Department of Biostatistics, Harvard T.H. Chan School of Public Health, Boston, Massachusetts, USA
[9]Institute of Social & Preventive Medicine, University of Bern, Bern, Switzerland
[10]Centre for Infectious Disease Epidemiology and Research, University of Cape Town Faculty of Health Sciences, Observatory, Western Cape, South Africa
[11]Population Health Sciences, Bristol Medical School, University of Bristol, Bristol, UK
[12]RTI Health Solutions Barcelona, Barcelona, Catalunya, Spain
[13]Department of Medicine, Northwestern University Feinberg School of Medicine, Chicago, Illinois, USA
[14]Department of Biostatistics, Boston University School of Public Health, Boston, Massachusetts, USA
[15]Clinical Epidemiology & Biostatistics Unit, Murdoch Children's Research Institute, Melbourne, Victoria, Australia
[16]Department of Paediatrics, The University of Melbourne, Melbourne, Victoria, Australia
[17]School of Population Health, Faculty of Medicine and Health, UNSW Sydney, Sydney, New South Wales, Australia
[18]Division of Pharmacoepidemiology, Department of Medicine, Brigham & Women's Hospital, Harvard Medical School, Boston, Massachusetts, USA
[19]Department of Population Health Sciences, University of Bristol, Bristol, UK
[20]NIHR Bristol Biomedical Research Centre, Bristol, UK
[21]Health Data Research UK South-West, Bristol, UK
[22]Department of Public Health and Epidemiology, RCSI University of Medicine and Health Sciences, Dublin, Ireland
[23]Department of Biostatistics, Johns Hopkins Bloomberg School of Public Health, Baltimore, Maryland, USA
[24]University of Exeter Medical School, Exeter, UK
[25]EMEA Methods and Evidence Generation, IQVIA London, London, UK

**Acknowledgements** We acknowledge Nia Roberts, outreach librarian and information specialist at the University of Oxford for assistance designing the literature search.

**Contributors** HJH, AGC, MDJ, HL, JHM conceived the idea for the project protocol. All authors contributed to the design and methodology of the project protocol. HJH and AGC wrote the first draft of the manuscript. MAH, SAS, IJD, BAD, XG-A, ME, RMG, NI, SL, MM-B, S-AP, SS, JS, MKS, EAS provided feedback, revised the manuscript and have read and approved the final version.

**Funding** There was no specific funding for this study. HJH was supported by an Australian National Health and Medical Research Council (NHMRC) Postgraduate Scholarship, a PhD Top-Up Scholarship from Neuroscience Research Australia, and was a Neuroscience Research Australia PhD Pearl sponsored by Sandra Salteri AO. AGC was supported by an Australian NHMRC Investigator Grant (ID 2010088). MM-B was supported by an Australian NHMRC Investigator Grant (ID 2009572). JHM was supported by an Australian NHMRC Investigator Grant (ID 2010128). BAD was supported by a grant from the National Institutes of Health (R00 CA248335). ME was supported by grants from the National Institutes of Health (R01 AI152772-01, 5U01-AI069924-05) and the Swiss National Science Foundation (32FP30-174281). NI was supported by grants from the National Institute for Health and Care Research (HDRUK2022.0313) and the UK Office for National Statistics (2002563). MAH was supported by NIH grant R37 AI102634.

**Competing interests** None declared.

**Patient and public involvement** Patients and/or the public were not involved in the design, or conduct, or reporting, or dissemination plans of this research.

**Patient consent for publication** Not applicable.

**Provenance and peer review** Not commissioned; externally peer reviewed.

**ORCID iDs**
Harrison J Hansford http://orcid.org/0000-0002-5942-8509
Aidan G Cashin http://orcid.org/0000-0003-4190-7912
Matthias Egger http://orcid.org/0000-0001-7462-5132
Jonathan Sterne http://orcid.org/0000-0001-8496-6053

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
