## [Reviewer comments · BMJ Open]

ARTICLE DETAILS

TITLE (PROVISIONAL)	Development of the Transparent Reporting of Observational Studies Emulating a Target Trial (TARGET) Guideline
AUTHORS	Hansford, Harrison; Cashin, Aidan; Jones, Matthew; Swanson, Sonja; Islam, Nazrul; Dahabreh, Isa; Dickerman, Barbra; Egger, Matthias; Garcia-De-Albeniz, Xavier; Golub, Robert; Lodi, Sara; Moreno-Betancur, Margarita; Pearson, Sallie-Anne; Schneeweiss, Sebastian; Sterne, Jonathan; Sharp, Melissa; Stuart, Elizabeth A; Hernan, M; Lee, Hopin; McAuley, James

VERSION 1 – REVIEW

REVIEWER	Hassan, Muhammad Muneeb The Islamia University of Bahawalpur Pakistan, Statistics
REVIEW RETURNED	27-Apr-2023

GENERAL COMMENTS	Some minor changes are required, which I already mentioned in the comments file. Development of the Transparent Reporting of Observational Studies Emulating a Target Trial (TARGET) Guideline The concept of this project is very interesting, and the authors have used some useful methodologies to work on the domain of targeted trial by using Observational studies. But there are some minor revisions that need to be addressed before this paper can be published. Abstract Abstract should be more concise; carefully consider potential biases and limitations throughout the process. Reporting guidelines should be developed through a transparent and rigorous process that involves input from multiple stakeholders. Failure to follow such a process may result in guidelines that are incomplete, biased, or not widely accepted. Even the best reporting guidelines are unlikely to be effective if they are not widely disseminated and adopted by the scientific community. It is important to have a plan for disseminating the guideline and encouraging its adoption to ensure that it has an impact on the quality of observational studies.
--

Introduction

The introduction does not clearly explain the rationale behind the study or the hypothesis of the intended work. I think it should revise in all aspects with recent references. Authors are encouraged to read and cite the article to make the analysis process for building different models. <https://doi.org/10.1080/07853890.2022.2160491>

Methods

The method section should be in the stream line in continuity; separate all the stages you use in this section from all other information at the start. I think the whole method section should be revised.

1. It is unclear whether the search will include all relevant databases. While you states that the search will include Medline, EMBASE, PsycINFO, and the Science Citation Index, there may be other databases that are relevant to the topic.
2. The exclusion criteria seem to be too restrictive. Excluding studies that are not conducted in humans or in the field of medicine and health may eliminate some important studies that provide relevant information. It may also introduce bias by excluding studies that are conducted in non-human populations or in other fields but have potential applicability to medicine and health.
3. The search terms may not be comprehensive enough to capture all relevant studies. While the authors have used sensitive search terms, including emulat*, target trial, observational data, real-world data, comparative effectiveness, and causal inference, there may be other relevant terms that are not included in the search.
4. It is unclear how the authors will address potential language bias. Excluding studies not written in English may limit the scope of the review and exclude important studies that are published in other languages.
5. There is no mention of grey literature or unpublished

	studies. The authors may miss some relevant studies if they do not search for grey literature or include unpublished studies in their review. 6. The use of a standardized data extraction form and the piloting of the form with a selection of included studies are appropriate methods to ensure consistency and accuracy in the extraction process. The plan also addresses potential disagreements between reviewers through discussion or consultation with a third reviewer. However, it is worth noting that the effectiveness of the data extraction process depends on the quality and completeness of the information provided in the included studies. If important information is missing or unclear in the studies, the data extraction process may be challenging or may not provide accurate results. Discussion In the discussion section, it is not well stated how useful this research can be in diagnosis and what the interpreters are. It should be more elaborate in detail.
--	--

REVIEWER	Wang, Xiaofeng The Cleveland Clinic, Quantitative Health Sciences
REVIEW RETURNED	12-Jun-2023

GENERAL COMMENTS	Target trial emulation has drawn great attention in the past few years. However, there is currently no guidance on reporting a study designed to emulate a target trial. This protocol describes the methods that the authors plan to develop the Transparent reporting of observational studies emulating a target trial (TARGET) reporting guideline. This protocol paper is well-planned and clearly written. The proposed five steps follow the recommendations for the development of health research reporting guidelines. Specific comments: 1. Page 15, line 238 - line 239: "Continuous variables will be summarised using mean and standard deviation, and median and interquartile range" could rewrite as "Continuous variables will be summarised using mean and standard deviation, or median and interquartile range, as appropriate". 2. Page 15, line 239 "We will analyse the free-text responses from
---

	the first and second surveys using an inductive approach, in which we will use reflexive thematic analysis..." Many readers may not be familiar with your inductive approach. Please provide the related references for your approach. Adding one or two sentences about the details of how you analyze the free-text response might be useful.
--	---

VERSION 1 – AUTHOR RESPONSE

Reviewer #1:

Development of the Transparent Reporting of Observational Studies Emulating a Target Trial (TARGET) Guideline

The concept of this project is very interesting, and the authors have used some useful methodologies to work on the domain of targeted trial by using Observational studies. But there are some minor revisions that need to be addressed before this paper can be published.

Abstract

3. **Abstract should be more concise; carefully consider potential biases and limitations throughout the process. Reporting guidelines should be developed through a transparent and rigorous process that involves input from multiple stakeholders. Failure to follow such a process may result in guidelines that are incomplete, biased, or not widely accepted. Even the best reporting guidelines are unlikely to be effective if they are not widely disseminated and adopted by the scientific community. It is important to have a plan for disseminating the guideline and encouraging its adoption to ensure that it has an impact on the quality of observational studies.**

- We have added more detail about the process by which we will be developing the guidance (following recommended guidance for developers of health research reporting guidelines from Moher et al., 2010 - 10.1371/journal.pmed.1000217). We have also added information about how we will ensure the likelihood of the guideline will be adopted widely.

- o "Background

Observational studies are increasingly used to inform health decision-making when randomised trials are not feasible, ethical, or timely. The target trial approach provides a framework to help minimise common biases in observational studies that aim to estimate the causal effect of interventions. Incomplete reporting of studies using the target trial framework limits the ability for clinicians, researchers, patients, and other decision-makers to appraise, synthesise, and interpret findings to inform clinical and public health practice and policy. This paper describes the planned methods to develop the Transparent reporting of observational studies emulating a target trial (TARGET) reporting guideline.

Methods/design

The TARGET reporting guideline will be developed in five stages following recommended guidance. The first stage will identify current target trial reporting practices by systematically reviewing published studies that explicitly emulated a target trial. The second stage will identify and refine items to be considered for inclusion in the TARGET guideline by consulting content experts using two online surveys. The third stage will prioritise and consolidate key items to be included in the TARGET guideline at a consensus meeting of TARGET investigators. The

fourth stage will produce and pilot-test the TARGET guideline and explanation and elaboration document with all relevant stakeholders. The fifth stage will disseminate the TARGET guideline and resources via journals, conferences, and courses.

Ethics and Dissemination

Ethical approval for the survey to be conducted has been attained (HC220536). The TARGET guideline will be disseminated widely in partnership with stakeholders to maximise adoption and improve reporting of these studies."

Introduction

4. **The introduction does not clearly explain the rationale behind the study or the hypothesis of the intended work. I think it should revise in all aspects with recent references. Authors are encouraged to read and cite the article to make the analysis process for building different models.** <https://doi.org/10.1080/07853890.2022.2160491>
- a. We thank you for your suggestion, we explain the rationale for conducting out study in paragraph three of our introduction.
 - o "Application of the target trial framework requires the complete specification of the target trial protocol and its emulation (Figure 1).³ Hernán & Robins³ provide a template for specifying a target trial and its emulation; however, there is currently no detailed guidance on reporting a study designed to emulate a target trial. **Incomplete reporting of these studies limits the ability of clinicians, researchers, patients, and other decision-makers to appraise and synthesise findings or interpret them to inform clinical and public health practice and policy. A reporting guideline that expands upon the initial target trial emulation template³ is needed to provide authors with comprehensive recommendations on how to completely and transparently report a study emulating a target trial.**" Line 122 to 131
 - b. As our report describes the process we will undertake to develop a reporting guideline, we do not believe a hypothesis is appropriate.
 - c. We read with interest the reference you provided, however we do not believe it fits in our introduction.
 - d. We have included recent references throughout the second paragraph.
 - The value of using the target trial framework to design the analysis of observational studies has been recognised by international regulatory bodies in the field of medicine and health,¹⁰⁻¹⁴ and the framework underpins the widely-used ROBINS-I tool for assessing risk of bias in non-randomised studies of interventions.¹⁵ Studies that are explicit in using the target trial framework have been published with increasing frequency in leading general medical and specialty journals.¹⁶⁻²¹ Line 115 to 120
 - "Concato J, Stein P, Dal Pan GJ, et al. Randomized, observational, interventional, and real-world—What's in a name? *Pharmacoepidemiology and drug safety* 2020;29(11):1514-17.
 - 11. Agency EM. European medicines agencies network strategy to 2025. The Netherlands: Health of Medicines Agencies, 2020.
 - 12. Excellence NIfHaC. The NICE strategy 2021 to 2026, 2021.
 - 13. Health Canada. Optimizing the Use of Real World Evidence to Inform Regulatory Decision-Making: Government of Canada, 2019.
 - 14. Therapeutic Goods Administration. Real world evidence and patient reported outcomes in the regulatory context.

<https://www.tga.gov.au/review-real-world-evidence-and-patient-reported-outcomes>: Australian Government, Department of Health, 2021.

- 15. Sterne JA, Hernán MA, Reeves BC, et al. ROBINS-I: a tool for assessing risk of bias in non-randomised studies of interventions. *BMJ* 2016;355:i4919. doi: 10.1136/bmj.i4919
- 16. Garcia-Albeniz XH, J.: Bretthauer, M.: Hernan, M. A. Effectiveness of screening colonoscopy to prevent colorectal cancer among medicare beneficiaries aged 70 to 79 years: A prospective observational study. *Annals of Internal Medicine* 2017 doi: <http://dx.doi.org/10.7326/M16-0758>
- 17. Mahévas M, Tran V-T, Roumier M, et al. Clinical efficacy of hydroxychloroquine in patients with covid-19 pneumonia who require oxygen: observational comparative study using routine care data. *BMJ* 2020;369:m1844. doi: 10.1136/bmj.m1844
- 18. Kaura A, Sterne JAC, Trickey A, et al. Invasive versus non-invasive management of older patients with non-ST elevation myocardial infarction (SENIOR-NSTEMI): a cohort study based on routine clinical data. *The Lancet* 2020;396(10251):623-34. doi: 10.1016/S0140-6736(20)30930-2
- 19. Emilsson L, García-Albéniz X, Logan RW, et al. Examining Bias in Studies of Statin Treatment and Survival in Patients With Cancer. *JAMA Oncol* 2018;4(1):63-70. doi: 10.1001/jamaoncol.2017.2752
- 20. Chan You S, Krumholz HM, Suchard MA, et al. Comprehensive Comparative Effectiveness and Safety of First-Line β -Blocker Monotherapy in Hypertensive Patients: A Large-Scale Multicenter Observational Study. *Hypertension* 2021;77(5):1528-38. doi: 10.1161/hypertensionaha.120.16402 [published Online First: 20210329]
- 21. Caniglia EC, Robins JM, Cain LE, et al. Emulating a trial of joint dynamic strategies: An application to monitoring and treatment of HIV-positive individuals. *Stat Med* 2019;38(13):2428-46. doi: 10.1002/sim.8120 [published Online First: 20190318]"

Methods

5. The method section should be in the stream line in continuity; separate all the stages you use in this section from all other information at the start. I think the whole method section should be revised.

- a. We acknowledge that the format of this protocol is slightly different to most study protocols as we outline plans for five different stages. We have separated out each of these stages and briefly outline our planned processes for each stage in the order they will occur to be as streamlined as possible.

6. It is unclear whether the search will include all relevant databases. While you states that the search will include Medline, EMBASE, PsycINFO, and the Science Citation Index, there may be other databases that are relevant to the topic.

- We thank you for your suggestion. We acknowledge there may be other relevant databases that we may not have included, however, Medline and Embase index the vast majority of scientific, medical literature, and to ensure that we do not miss relevant studies, we will conduct forward citation tracking of key seminal articles, as well as include studies known to the authorship team.

- a. "The complete search strategy is in Supplementary Material 2. We will conduct forward citation tracking of selected seminal articles to maximise the

chance of retrieving all relevant articles.3,4,8,19,20 We will also include papers known to the authorship team.” Line 190 to 193

7. The exclusion criteria seem to be too restrictive. Excluding studies that are not conducted in humans or in the field of medicine and health may eliminate some important studies that provide relevant information. It may also introduce bias by excluding studies that are conducted in non-human populations or in other fields but have potential applicability to medicine and health.

- We thank you for your suggestion and agree that some studies may be excluded with our eligibility criteria. However, our intention is for the TARGET Guideline to primarily be relevant to studies conducted in humans and of medical or health interventions, therefore we politely decline to broaden our eligibility criteria to ensure we keep the studies included relevant to our overall goal.

8. The search terms may not be comprehensive enough to capture all relevant studies. While the authors have used sensitive search terms, including emulat*, target trial, observational data, real-world data, comparative effectiveness, and causal inference, there may be other relevant terms that are not included in the search.

- a. We have included our full search strategy in Supplementary Material 2 which was developed to be sensitive by a Librarian and information specialist at the University of Oxford. To further ensure all relevant studies are retrieved we have conducted forward citation tracking of seminal articles, and articles known to the expert authorship team.
 - o “The complete search strategy is in Supplementary Material 2. We will conduct forward citation tracking of selected seminal articles to maximise the chance of retrieving all relevant articles.3,4,8,19,20 We will also include papers known to the authorship team.” Line 190 to 193

9. It is unclear how the authors will address potential language bias. Excluding studies not written in English may limit the scope of the review and exclude important studies that are published in other languages.

- We acknowledge that excluding studies not written in English may limit the scope of the review, however, we do not have the resources to translate all documents, and given the nature of the data we are collecting from the studies (i.e., how aspects of the studies are reported), we decided in order to most validly extract the data we should limit our search to only studies written in English. We have added a comment on this point.

a. *“Outcomes of the systematic review*

The systematic review will provide evidence on reporting in studies explicitly emulating a target trial. We acknowledge that excluding studies not written in English and unpublished studies may cause potentially relevant articles to be excluded. The findings will inform the online surveys (Stage 2) and the consensus meeting (Stage 3). We will submit the findings of this review for publication and all data and code made publicly available.” Line 220 to 225

10. There is no mention of grey literature or unpublished studies. The authors may miss some relevant studies if they do not search for grey literature or include unpublished studies in their review.

- We agree and acknowledge this limitation. We decided to not include studies that were unpublished due to the impact that journal publication may have on reporting, therefore to create a homogenous sample we decided to restrict our search and eligibility criteria to only published studies.

a. *“Outcomes of the systematic review*

The systematic review will provide evidence on reporting in studies explicitly emulating a target trial. We acknowledge that excluding studies not written in English and unpublished studies may cause potentially relevant articles to be excluded. The findings will inform the online surveys (Stage 2) and the consensus meeting (Stage 3). We will submit the findings of this review for publication and all data and code made publicly available.” Line 220 to 225

Discussion

11. In the discussion section, it is not well stated how useful this research can be in diagnosis and what the interpreters are. It should be more elaborate in detail.

- As per the editors comment and journal policy for protocols we have removed the discussion.

Reviewer #2:

Target trial emulation has drawn great attention in the past few years. However, there is currently no guidance on reporting a study designed to emulate a target trial.

This protocol describes the methods that the authors plan to develop the Transparent reporting of observational studies emulating a target trial (TARGET) reporting guideline. This protocol paper is well-planned and clearly written. The proposed five steps follow the recommendations for the development of health research reporting guidelines.

Specific comments:

12. Page 15, line 238 - line 239: "Continuous variables will be summarised using mean and standard deviation, and median and interquartile range" could rewrite as "Continuous variables will be summarised using mean and standard deviation, or median and interquartile range, as appropriate".

- We have amended this section.
 - a. “Continuous variables will be summarised using mean and standard deviation, or median and interquartile range, as appropriate” Line 267 to 268

13. Page 15, line 239 "We will analyse the free-text responses from the first and second surveys using an inductive approach, in which we will use reflexive thematic analysis..." Many readers may not be familiar with your inductive approach. Please provide the related references for your approach. Adding one

or two sentences about the details of how you analyze the free-text response might be useful.

- We have added more detail and references to our description of our qualitative analysis
 - a. "We will analyse the free-text responses from the first and second surveys using an inductive approach,³⁰ in which we will use reflexive thematic³⁰ analysis to identify, organise and generate codes, and then identify themes found within the dataset. Briefly, inductive coding is a process pooling common ideas without trying to fit ideas/codes into a pre-existing framework." Line 271 to 272

VERSION 2 – REVIEW

REVIEWER	Hassan, Muhammad Muneeb The Islamia University of Bahawalpur Pakistan, Statistics
REVIEW RETURNED	27-Jul-2023

GENERAL COMMENTS	According to your observation, your efforts are going in the right direction in the Development of the Transparent Reporting of Observational Studies Emulating a Target Trial (TARGET) Guideline, but I am still not satisfied with the author’s justifications; it still requires more improvements. Abstract  1. "Observational studies are increasingly being used." Suggested rephrasing: "Observational studies are increasingly being used." 2. "when randomized trials are not feasible, ethical, or timely," rephrasing it to "when randomized trials are infeasible, unethical, or not timely." 3. "The target trial approach provides a framework to help minimize common biases." Capitalize "target trial" to maintain consistency. 4. "estimating the causal effect of interventions": Replace "estimating" with "estimation" for grammatical correctness. 5. "However, incomplete reporting of studies using the target trial framework limits the ability." rephrasing into "Incomplete reporting of studies utilizing the target trial framework limits the ability." 6. "for clinicians, researchers, patients, and other decision-makers to appraise, synthesize, and interpret findings to inform clinical and public health practice and policy." Rephrase to "for clinicians, researchers, patients, and decision-makers to appraise, synthesize, and interpret findings, thus informing clinical and public health practice and policy." 7. "Ethical approval for the survey to be conducted has been attained (HC220536)." - Mention the specific body or institution that granted the ethical approval.
--

8. Ambiguity: The phrase "maximize adoption" needs to be clarified as to what it means in the context of disseminating the TARGET guideline.
9. "emulated a target trial—capitalize "target trial" for consistency.
10. "by consulting content experts using two online surveys." Clarify whether the surveys are sequential or concurrent.
11. "consensus meeting of TARGET investigators: specify the nature of the consensus meeting (e.g., in-person, virtual).
12. "Produce and pilot-test the Target guideline and explanation and elaboration document." Consider rephrasing to "produce and pilot-test both the TARGET guideline and the explanation and elaboration document."

Introduction

1. Non-relevant information: Line 107 mentions the proposal of using observational data to mimic a randomised experiment in the mid-20th century. While this historical information might be interesting, it may not be directly relevant to the current topic of developing the TARGET reporting guideline.
2. Repeat information about the target trial framework and its two stages. This repetition could be consolidated to avoid redundancy.
3. Line 120 mentions the incomplete reporting of studies emulating a target trial, but the specific figure (Figure 1) mentioned earlier in the text, which is relevant to the specification and emulation of the target trial, is not cited here.
4. In line 131, the figure (Figure 1) is cited, but the citation style is not consistent with other references in the text.

The introduction does not clearly explain the rationale behind the study or the hypothesis of the intended work. Some sentences can be rephrased for improved clarity and readability. Carefully read and cite the recent articles:

- Zhao SS, Lyu H, Solomon DH, et al Improving rheumatoid arthritis comparative effectiveness research through causal inference principles: systematic review using a target trial emulation framework *Annals of the Rheumatic Diseases* 2020;**79**:883-890.
- Bigirimurame T, Hiu SKW, Teare MD, et al Current practices in studies applying the target trial emulation framework: a protocol for a systematic review *BMJ Open* 2023;**13**:e070963. doi: 10.1136/bmjopen-2022-070963

	 • https://doi.org/10.1080/23311916.2023.2224137 • https://doi.org/10.1080/07853890.2022.2160491 Databases, eligibility, and search terms/data analysis/ Outcomes of the systematic review etc  1. That's the future tense you're using, right? Please use the past tense. Fix it up 'round the article. Kindly make the necessary changes. Also make it fluent again. R-Codes You are saying that you analyzed the free-text responses from the first and second surveys. Using an inductive approach, in which you used reflexive thematic analysis to identify, organize, and generate codes, and then identify themes found within the dataset, that's good. I need your R codes for verification of the results that identify themes found within the dataset.
--	---

REVIEWER	Wang, Xiaofeng The Cleveland Clinic, Quantitative Health Sciences
REVIEW RETURNED	02-Aug-2023

GENERAL COMMENTS	The authors have addressed all of my comments.
--

VERSION 2 – AUTHOR RESPONSE

REVIEWER 1.

Abstract

1. **"Observational studies are increasingly being used." Suggested rephrasing: "Observational studies are increasingly being used."**

Response: This suggestion is the same as the current text, we have kept the text as suggested.

2. **"when randomized trials are not feasible, ethical, or timely," rephrasing it to "when randomized trials are infeasible, unethical, or not timely."**

Response: We thank the reviewer for the suggestion, however we have decided to keep our original phrasing because it is more concise.

- 3. "The target trial approach provides a framework to help minimize common biases." Capitalize "target trial" to maintain consistency.**

Response: As 'target trial' is not a proper noun we refrain from capitalising it to maintain consistency in the field and the seminal paper from Hernán and Robins (2016).

- 4. "estimating the causal effect of interventions": Replace "estimating" with "estimation" for grammatical correctness.**

Response: Estimation would not be grammatically correct, therefore we have decided to keep our original phrasing.

- 5. "However, incomplete reporting of studies using the target trial framework limits the ability." rephrasing into "Incomplete reporting of studies utilizing the target trial framework limits the ability."**

Response: We have not changed 'using' to 'utilizing' as we believe 'using' is easier to understand for readers.

- 6. "for clinicians, researchers, patients, and other decision-makers to appraise, synthesize, and interpret findings to inform clinical and public health practice and policy." Rephrase to "for clinicians, researchers, patients, and decision-makers to appraise, synthesize, and interpret findings, thus informing clinical and public health practice and policy."**

Response: The suggested change is not grammatically correct therefore we have kept our original phrasing.

- 7. "Ethical approval for the survey to be conducted has been attained (HC220536)." - Mention the specific body or institution that granted the ethical approval.**

Response: We have outlined the ethical review body in the main text, however, due to the word limit are unable to include this information in the abstract.

Line 215 to 216: Ethical approval has been obtained for the online surveys from the University of New South Wales Human Research Ethics Committee (HC220536).

8. Ambiguity: The phrase "maximize adoption" needs to be clarified as to what it means in the context of disseminating the TARGET guideline.

Response: We have information in the main text that outlines what we mean by 'maximize adoption,' (included below) however, we are unable to expand upon this in the abstract due to the word limit.

Line 322 to 335: Stage 5 – Guideline implementation

The goal of the final stage of guideline development is to maximise reach and use of the TARGET guideline. The TARGET working group will guide the dissemination strategy with advice from consensus meeting participants. We aim to publish the TARGET guideline and the explanation and elaboration document and disseminate the findings through traditional and social media. We will engage journal editors and funding agencies to encourage TARGET guideline endorsement alongside other published reporting guidance. We will publicly host the TARGET guideline and explanation and elaboration paper, and any other relevant material on a TARGET website. We will index the guideline on the Enhancing the QUALity and Transparency Of health Research (EQUATOR) Network website.³²
³³ We will create online resources including infographics, blog posts and podcasts, which will be available on the TARGET website. We will share the TARGET guideline with authors in the field, and at relevant scientific conferences and methodological courses.

9. "emulated a target trial—capitalize "target trial" for consistency.

Response: As per response #3, we refrain from capitalising 'target trial'.

10. "by consulting content experts using two online surveys." Clarify whether the surveys are sequential or concurrent.

Response: We have clarified that the surveys are sequential

Lines 72 to 73: The second stage will identify and refine items to be considered for inclusion in the TARGET guideline by consulting content experts using sequential online surveys.

11. "consensus meeting of TARGET investigators: specify the nature of the consensus meeting (e.g., in-person, virtual).

Response: We have clarified the nature of our consensus meeting.

Lines 74 to 75: The third stage will prioritise and consolidate key items to be included in the TARGET guideline at an in-person consensus meeting of TARGET investigators.

- 12. "Produce and pilot-test the Target guideline and explanation and elaboration document." Consider rephrasing to "produce and pilot-test both the TARGET guideline and the explanation and elaboration document."**

Response: We have adapted the phrase as suggested.

Lines 75 to 77: "The fourth stage will produce and pilot-test both the TARGET guideline and explanation and elaboration document with relevant stakeholders."

Introduction

- 13. Non-relevant information: Line 107 mentions the proposal of using observational data to mimic a randomised experiment in the mid-20th century. While this historical information might be interesting, it may not be directly relevant to the current topic of developing the TARGET reporting guideline.**

Response: We believe outlining the history of the target trial framework is important to highlight that we are not suggesting a reporting guideline for a new framework. We have kept the text as currently written.

- 14. Repeat information about the target trial framework and its two stages. This repetition could be consolidated to avoid redundancy.**

Response: We have included this information deliberately to emphasise the two stages as from our previous work we are aware that researchers do not separate the stages.

- 15. Line 120 mentions the incomplete reporting of studies emulating a target trial, but the specific figure (Figure 1) mentioned earlier in the text, which is relevant to the specification and emulation of the target trial, is not cited here.**

Response: We had already referenced Figure 1 as recommended, below is the text we had included.

Line 120 to 121: Application of the target trial framework requires the complete specification of the target trial protocol and its emulation (Figure 1).³

- 16. In line 131, the figure (Figure 1) is cited, but the citation style is not consistent with other references in the text.**

Response: This text is placeholder text as a recommendation for the journal to insert Figure 1 into the text at this point, not a formal citation.

17. The introduction does not clearly explain the rationale behind the study or the hypothesis of the intended work. Some sentences can be rephrased for improved clarity and readability. Carefully read and cite the recent articles:
- Zhao SS, Lyu H, Solomon DH, et al Improving rheumatoid arthritis comparative effectiveness research through causal inference principles: systematic review using a target trial emulation framework *Annals of the Rheumatic Diseases* 2020;79:883-890.
 - Bigirumurame T, Hiu SKW, Teare MD, et al Current practices in studies applying the target trial emulation framework: a protocol for a systematic review *BMJ Open* 2023;13:e070963. doi: 10.1136/bmjopen-2022-070963
 - <https://doi.org/10.1080/23311916.2023.2224137>
 - <https://doi.org/10.1080/07853890.2022.2160491>

Databases, eligibility, and search terms/data analysis/ Outcomes of the systematic review etc

Response: We have read the papers and have included several citations in our introduction of recent systematic reviews highlighting the use of the target trial framework. (Scola, G., et al. Implementation of the trial emulation approach in medical research: a scoping review. *BMC Medical Research Methodology* 23, 186 (2023). & Zuo, H., Yu, L., Campbell, S.M., Yamamoto, S.S. & Yuan, Y. The implementation of target trial emulation for causal inference: a scoping review. *J Clin Epidemiol* (2023).)

Lines 115 to 118: Studies that are explicit in using the target trial framework have been published with increasing frequency in leading general medical and specialty journals.¹⁶⁻²³

18. That's the future tense you're using, right? Please use the past tense. Fix it up 'round the article. Kindly make the necessary changes. Also make it fluent again.

Response: We have left manuscript as being written in future tense as this is a protocol that has not been completed yet.

19. R-Codes

You are saying that you analyzed the free-text responses from the first and second surveys. Using an inductive approach, in which you used reflexive thematic analysis to identify, organize, and generate codes, and then identify themes found within the dataset, that's good.

I need your R codes for verification of the results that identify themes found within the dataset.

Response: We have not conducted the study yet as this is a protocol therefore cannot provide our codes at this point.

VERSION 3 – REVIEW

REVIEWER	Hassan, Muhammad Muneeb The Islamia University of Bahawalpur Pakistan, Statistics
REVIEW RETURNED	24-Aug-2023
GENERAL COMMENTS	I hope this message finds you well. I would like to express my heartfelt appreciation for your diligent efforts in revising the study titled "Development of the Transparent Reporting of Observational Studies Emulating a Target Trial (TARGET) Guideline." Author willingness to address the comments and make necessary changes showcases their commitment to producing high-quality research. Authors respond well according to my observation and expectation.